# Hepatic Ly6C^Lo^ Non-Classical Monocytes Have Increased Nr4a1 (Nur77) in Murine Biliary Atresia

**DOI:** 10.3390/jcm11185290

**Published:** 2022-09-08

**Authors:** Sarah Mohamedaly, Claire S. Levy, Cathrine Korsholm, Anas Alkhani, Katherine Rosenberg, Judith F. Ashouri, Amar Nijagal

**Affiliations:** 1Department of Surgery, University of California, San Francisco, CA 94143, USA; 2Liver Center, University of California, San Francisco, CA 94143, USA; 3Department of Comparative Pediatrics and Nutrition, University of Copenhagen, 1870 Frederiksberg, Denmark; 4Rosalind Russell and Ephraim P. Engleman Rheumatology Research Center, Department of Medicine, University of California, San Francisco, CA 94143, USA; 5The Pediatric Liver Center, UCSF Benioff Children’s Hospital, San Francisco, CA 94143, USA; 6Eli and Edythe Broad Center of Regeneration Medicine, University of California, San Francisco, CA 94143, USA

**Keywords:** perinatal liver inflammation, innate immune system, monocytes, biliary atresia, cholangiopathy

## Abstract

Biliary atresia (BA) is a rapidly progressive perinatal inflammatory disease, resulting in liver failure. Hepatic Ly6C^Lo^ non-classical monocytes promote the resolution of perinatal liver inflammation during rhesus rotavirus-mediated (RRV) BA in mice. In this study, we aim to investigate the effects of inflammation on the transcription factor Nr4a1, a known regulator of non-classical monocytes. Nr4a1-GFP reporter mice were injected with PBS for control or RRV within 24 h of delivery to induce perinatal liver inflammation. GFP expression on myeloid immune populations in the liver and bone marrow (BM) was quantified 3 and 14 days after injection using flow cytometry. Statistical significance was determined using a student’s *t*-test and ANOVA, with a *p*-value < 0.05 for significance. Our results demonstrate that non-classical monocytes in the neonatal liver exhibit the highest mean fluorescence intensity (MFI) of Nr4a1 (Ly6C^Lo^ MFI 6344 vs. neutrophils 3611 *p* < 0.001; macrophages 2782; *p* < 0.001; and Ly6C^Hi^ classical monocytes 4485; *p* < 0.0002). During inflammation, hepatic Ly6C^Lo^ non-classical monocytes showed a significant increase in Nr4a1 expression intensity from 6344 to 7600 (*p* = 0.012), while Nr4a1 expression remained unchanged on the other myeloid populations. These findings highlight the potential of using Nr4a1 as a regulator of neonatal hepatic Ly6C^Lo^ non-classical monocytes to mitigate perinatal liver inflammation.

## 1. Introduction

Biliary atresia (BA) is a perinatal hepatic inflammatory disease that results in rapid obliteration of the biliary tree, leading to cirrhosis and requiring liver transplantation. The etiology of BA is not fully understood, but multiple factors including genetic predisposition, immune dysregulation, toxins, and infections are thought to ultimately result in an inflammatory cascade within the liver [1]. Although a well-controlled acute inflammatory response is essential for the resolution of tissue injury, a dysregulated inflammatory response can lead to the development of pathological inflammation and devastating long-term consequences [2]. Thus, understanding how perinatal hepatic inflammation is propagated is an essential prerequisite for developing therapeutic targets that can interrupt this inflammatory cascade and alleviate the need for liver transplantation in patients with BA.

The innate immune system, particularly through the actions of monocytes, plays a crucial role in the initiation and resolution of inflammation [3,4,5,6]. There are two main subtypes of monocytes: Ly6C^Hi^ classical monocytes are pro-inflammatory [3,7,8], and Ly6C^Lo^ non-classical monocytes are pro-reparative [9,10,11,12]. In a murine model of BA, we have demonstrated that the relative abundance of Ly6C^Lo^ non-classical monocytes promotes the resolution of periportal liver inflammation and confers protection to liver disease in neonates [12]. However, the development and differentiation of Ly6C^Lo^ non-classical monocytes in neonatal mice are not well understood. In adult murine bone marrow, the transcription factor Nr4a1 has been shown to play a critical role in mediating the differentiation and survival of Ly6C^Lo^ non-classical monocytes [13]. Nr4a1 is an orphan nuclear receptor and a member of the Nr4a family of intracellular transcription factors [14,15]. The Nr4a1 family is widely expressed across various tissues, and its diverse scope of actions includes DNA repair, cell proliferation, differentiation, apoptosis, migration, metabolism, and inflammation [14,16]. Such diverse roles stem from the fact that Nr4a1 activity is tissue- and environment-specific [17,18,19].

Given the importance of Ly6C^Lo^ non-classical monocytes in promoting the resolution of perinatal liver inflammation, and our current understanding of Nr4a1 as a monocyte regulator in the adult murine bone marrow, we sought to investigate the effects of inflammation on Nr4a1 expression in a murine model of BA. Understanding the factors that regulate non-classical monocytes during perinatal inflammation is particularly important, because the early dysregulation of monocyte development can contribute to detrimental sequelae of BA.

## 2. Materials and Methods

**Mice.** BALB/c wildtype mice were obtained from the National Cancer Institute (Wilmington, MA) and Jackson labs (Bar Harbor, ME). Nr4a1-GFP reporter mice were originally described by Zikherman et al. [20] and were backcrossed onto the BALB/c background for >12 generations. All mouse experiments were approved by the UCSF Institutional Animal Care and Use Committee, and animals received humane care in accordance with the criteria outlined in the Guide for the Care and Use of Laboratory Animals. Mice were euthanized by decapitation (pups ≥ 7 days old) or carbon dioxide inhalation (pups >  7 days of age).

**Creation of single-cell suspensions from fetal and neonatal livers.** The liver and bone marrow were isolated from euthanized mice. To create single-cell suspensions, livers were weighed upon extraction and washed in cold PBS over ice for 5 min. Using sterile fine surgical scissors, livers were cut into small pieces to facilitate mechanical dissociation in cold PBS (for pups at 3 days of life) and in 2.5 mg/mL Liberase (Roche, Indianapolis, IN, USA, 05401119001) homogenization buffer (for pups at 14 days of life). Once the liver was completely homogenized, the cell suspension was passed through a 100 µm cell strainer to remove any undigested tissue or debris and to create SCS. Neonatal bone marrow was extracted by harvesting the sacral spine, pelvic bone, both femurs and tibias from each pup and washing them in cold PBS over ice for 5 min followed by crushing the bones using a pestle and mortar and washing the cells using cold PBS once the bones were completely crushed. Washed cells were then passed through a 100 µm cell strainer to remove any undigested bone or debris and to create an SCS. Both the liver and BM SCS were then incubated in 1 mL of ACK Lysis buffer (Gibco, Waltham, MA, USA, A10492-01) over ice to remove red blood cells, followed by manual cell counting to determine cell count and viability. Cells were also stained using Ghost Live/Dead stain (Tonbo, San Diego, CA, USA, 13-0870-T100) to determine cell viability in analysis.

**Flow cytometry.** Liver and bone marrow single-cell suspensions were stained using the following antibodies: Cd11c, clone N418 (Biolegend, San Diego, CA, USA, 117339); Ly6C, clone HK1.4 (Biolegend, 128035); MHCII, clone M5/114.15.2 (eBioscience, Waltham, MA, USA, 48-5321-82); Cd45, clone 30-F11 (eBioscience, 56-0451-82); Cd11b, clone M1/70 (eBioscience, 47-0112-82); Cd64, clone X54-5/7.1 (BD Biosciences, Franklin Lakes, NJ, USA, 741024); Ly6g, clone 1A8 (BD Biosciences, 560601); Fc block Cd16/Cd32, clone 2.4G2 (BD Biosciences, 553142); Ly6g, RB6-8c5 (Tonbo, San Diego, CA, USA, 60-5931); Ghost (Tonbo, 13-0870-T100). Flow cytometric data were acquired on a BD LSRII Fortessa X20 and analyzed using FlowJo (v10.8.1 Franklin Lakes, NJ, USA).

**Postnatal model of perinatal inflammation.** Rhesus rotavirus (RRV) and Cercopithecus aethiops kidney epithelial (MA104) cells were obtained from Dr. Henry Greenberg (Stanford University, Palo Alto, CA, USA). The virus was grown and titered in MA104 cells. To induce postnatal hepatic inflammation, P0 pups were injected intraperitoneally with 1.5 × 10^6^ focus-forming units (ffu) 24 h after delivery. Controls were injected with PBS using the same technique.

**Data analysis.** To compare the two groups, the unpaired t-test with Welch’s correction, the non-parametric Mann–Whitney test, and the Chi-square test were used, respectively, for normally distributed variables, non-normally distributed variables, and proportions. A one-way ANOVA with Tukey’s multiple comparisons test was used to compare multiple groups. Myeloid populations after Ab-mediated depletion were quantified by dividing the number of live cells of interest by organ weight. Graphpad Prism 9.0 (San Diego, CA, USA) was used to generate graphs and perform statistical analysis.

## 3. Results

**Ly6C^Lo^ non-classical monocytes reside in the neonatal liver.** We quantified the proportion of mature myeloid populations in the neonatal liver and bone marrow under homeostatic conditions. Using flow cytometry, we identified neutrophils (Ly6G+, MHCII−), macrophages (CD64+, CD11b+/−), and monocytes (CD64− CD11b+). We further separated monocytes into Ly6C^Hi^ classical and Ly6C^Lo^ non-classical monocytes (Figure 1a). Monocytes, particularly Ly6C^Lo^ non-classical monocytes, were abundant in the liver but not in the bone marrow (Figure 1b). This finding was similar to that derived in our previous work, demonstrating the relative abundance of Ly6C^Lo^ non-classical monocytes in the late-gestation fetus [12].

In P3 pups, monocytes comprised ~8% of all CD45+ leukocytes in the liver, compared to ~6% in the bone marrow (*p* = 0.018). Ly6C^Lo^ non-classical monocytes were the predominant monocyte subset in the liver (5.62% vs. 3.76% for bone marrow, *p* = 0.001) (Figure 1b). Higher proportions of neutrophils were seen in the bone marrow than in the liver (69.78% vs. 53.43%, *p* = 0.002), whereas macrophages comprised ~12% in both organs. In juvenile mice, monocytes no longer predominated in the liver at P14; instead, these were seen in higher proportions in the bone marrow (4.59% vs. liver 3.1%, *p* = 0.01) (Figure 1c). Ly6C^Hi^ classical monocytes were seen in higher proportions in the bone marrow (2.11% vs. liver 0.58%, *p* < 0.0001), whereas Ly6C^Lo^ non-classical monocytes comprised ~2.5% of all leukocytes in both organs at this timepoint (Figure 1c). Monocytes overall, and specifically Ly6C^Lo^ non-classical monocytes, were present in higher proportions in the P3 liver than in the bone marrow, and made up a smaller proportion of total leukocytes in P14 mice than in neonates (Figure 1c).

**Nr4a1 expression is highest in Ly6C^Lo^ non-classical monocytes compared to other myeloid populations in the neonatal liver.** We characterized Nr4a1 expression in mature myeloid populations under homeostatic conditions in the neonatal liver and bone marrow. To quantify the expression of Nr4a1, we used a reporter mouse in which the expression of enhanced green fluorescent protein (GFP) is under the control of the Nr4a1 regulatory region [20]. We observed that high proportions of all myeloid populations expressed Nr4a1 (%GFP-positive cells) in both the neonatal liver and the bone marrow (Figure 2a–d). In particular, significantly higher proportions of neutrophils, macrophages, and Ly6C^Hi^ classical monocytes in the neonatal bone marrow expressed Nr4a1 compared to the neonatal liver (Neutrophils: 80.03% vs. 66.52%; *p* = 0.02. Macrophages: 53.46% vs. 33.83%; *p* < 0.001. Ly6C^Hi^ classical monocytes: 66.13% vs. 56.51%; *p* = 0.03) (Figure 2a–c). In contrast, relatively equal proportions of Ly6C^Lo^ non-classical monocytes expressed Nr4a1 in both the neonatal bone marrow and liver (66.60% vs. 68.42%; *p* = 0.67) (Figure 2d).

Although Nr4a1 was expressed in moderately high proportions by neutrophils (Figure 2a), macrophages (Figure 2b) and monocyte subsets (Figure 2c,d), we evaluated the mean fluorescent intensity of Nr4a1 to quantify the level of Nr4a1 expression. Ly6C^Lo^ non-classical monocytes had significantly higher expression of Nr4a1 compared to populations in both the liver and the bone marrow (Figure 2e,f). Among the myeloid populations in the neonatal liver, Ly6C^Lo^ non-classical monocytes showed the highest intensity of Nr4a1 expression compared to neutrophils (MFI 6344 vs. 3611; *p* < 0.001), macrophages (MFI 6344 vs. 2782; *p* < 0.001), and Ly6C^Hi^ classical monocytes (MFI 6344 vs. 4485; *p* < 0.0002) (Figure 2e). Ly6C^Lo^ non-classical monocytes also exhibited the highest intensity of Nr4a1 expression in the neonatal bone marrow compared to other myeloid populations (Figure 2f); however, the Nr4a1 expression was significantly higher in liver Ly6C^Lo^ non-classical monocytes than in bone marrow Ly6C^Lo^ non-classical monocytes (MFI 6344 vs. 5338; *p* = 0.009) (Figure 2d). These results demonstrate that although Nr4a1 was present in high proportions in most myeloid populations, the expression was highest in Ly6C^Lo^ non-classical monocytes in the neonatal liver.

**Perinatal liver inflammation leads to increased expression of Nr4a1 in hepatic Ly6C^Lo^ non-classical monocytes.** Having shown that Nr4a1 expression is environment-specific, we next quantified the proportion of mature myeloid populations, and characterized their Nr4a1 expression in the setting of inflammation in the neonatal bone marrow (Figure 3) and liver (Figure 4). We used the well-established murine model of BA, which involves the infection of neonatal pups with Rhesus rotavirus. The injection of RRV during the first 24 h of life results in weight loss, jaundice, and death in the majority of neonatal pups by 21 days [12,21,22,23,24]. The characteristic histologic findings after RRV infection include periportal inflammation, which is apparent within 3 days. Our thorough analysis of the immune cells involved in the pathogenesis of RRV-mediated inflammation has demonstrated a limited role for the adaptive immune system and type 2 immunity in the pathogenesis of RRV, and has implicated Ly6C^Lo^ non-classical monocytes in the resolution of murine BA [12].

In the neonatal bone marrow, the neutrophil population decreased during inflammation (WT 67.44% vs. RRV 46.49%; *p* = 0.004), whereas other myeloid populations remained unchanged (Figure 3a,b). Nr4a1 expression remained stable in neutrophils (Figure 3c), decreased in macrophages (WT 4416 vs. RRV 3739; *p* = 0.04) (Figure 3d) and Ly6C^Lo^ non-classical monocytes (WT 5338 vs. RRV 4089; *p* = 0.003) (Figure 3f), and increased in Ly6C^Hi^ classical monocytes (MFI WT 4575 vs. RRV 6259; *p* = 0.0002) (Figure 3e).

In the neonatal liver, the levels of hepatic macrophages decreased during inflammation (WT 11.76% vs. RRV 6.53%; *p* < 0.0001), as did Ly6C^Lo^ non-classical monocytes (WT 4.96% vs. RRV 3.12%; *p* = 0.037) (Figure 4a–c), but the proportions of hepatic neutrophils and Ly6C^Hi^ classical monocytes were unchanged (Figure 4a). Despite these decreases in hepatic macrophages and Ly6C^Lo^ non-classical monocytes, inflammatory conditions resulted in an increase in Nr4a1 expression in those myeloid populations. The proportion of macrophages expressing Nr4a1 increased to 45.73% (WT 33.38%, *p* = 0.0078), and the intensity of Nr4a1 expression also increased from 2848 to 3752 (*p* = 0.013) (Figure 4e). Approximately 70% of all hepatic Ly6C^Lo^ non-classical monocytes expressed Nr4a1 under homeostatic and inflammatory conditions, but the intensity of Nr4a1 expression increased significantly from 6344 to 7600 during inflammation (*p* = 0.012) (Figure 4g). Nr4a1 expression remained unchanged in hepatic neutrophils and Ly6C^Hi^ classical monocytes Figure 4d,f).

## 4. Discussion

In this study, we hypothesized that Nr4a1 is important for Ly6C^Lo^ non-classical monocyte differentiation, and is therefore a key regulator of perinatal liver inflammation. Our results demonstrate that in the liver and bone marrow, Nr4a1 expression was highest among hepatic neonatal Ly6C^Lo^ non-classical monocytes compared to neutrophils, macrophages, and Ly6C^Hi^ classical monocytes. In a murine model of BA, although the levels of hepatic Ly6C^Lo^ non-classical monocytes decreased, their expression of Nr4a1 increased significantly. However, in the bone marrow, Nr4a1 expression decreased significantly during inflammation, but increased in Ly6C^Hi^ classical monocytes. Collectively, these results support the idea that Nr4a1 is an important transcription factor for the pro-reparative response mediated by hepatic Ly6C^Lo^ non-classical monocytes during inflammation in murine BA.

Ly6C^Lo^ non-classical monocytes have been shown to promote a pro-reparative response in the setting of inflammation, and specifically promote the resolution of periportal liver inflammation to confer protection to liver disease in murine neonates [9,10,11,12]. In our study, we demonstrate that the liver acts as a reservoir for Ly6C^Lo^ non-classical monocytes in neonatal mice before the physiologic homing process of hematopoietic cells to the bone marrow. While the homing process is a well-described phenomenon, the normal physiology and development of the murine hematopoietic system shortly before and after birth are not yet fully elucidated [25,26,27]. Hematopoietic cells have been examined extensively in the murine fetus [28,29,30]; however, mature cells at later stages of development, particularly in perinatal pups, have not been examined as thoroughly [31,32]. Hematopoietic cells are believed to migrate to the fetal liver at approximately day 10 post-coitus, and at or near birth, cells then migrate from the liver to the bone marrow, where they remain throughout the animal’s adult life [25,26,27]. Our results demonstrate that by postnatal day 14, the myeloid populations predominately reside in the bone marrow, and the liver no longer maintains a significant proportion of monocytes. Not only does this finding clarify the timing of the homing process from the liver to the bone marrow, but it also highlights that the neonatal liver initially maintains a high proportion of the pro-reparative Ly6C^Lo^ non-classical monocytes, raising the possibility that these cells may confer resistance to perinatal injury.

Previous studies have demonstrated the importance of Nr4a1 in the differentiation and survival of the pro-reparative Ly6C^Lo^ non-classical monocytes [13,33]. Monocyte subsets arise from a common monocyte progenitor (cMoP) in the adult murine bone marrow, but the intricacies of this development remain poorly understood due to the challenges of lineage tracing [33]. The most accepted hypothesis currently supports the idea that cMoPs give rise to Ly6C^Hi^ classical monocytes, which can then differentiate into Ly6C^Lo^ non-classical monocytes [33,34,35,36,37,38]. Multiple transcription factors such as PU.1, C/EBP-β and IRF8 have been implicated at various stages of monocyte development [39]; however, Nr4a1 has been most notably shown to regulate the differentiation of Ly6C^Lo^ non-classical monocytes in the adult murine bone marrow [13]. The relationship between Nr4a1 and Ly6C^Lo^ non-classical monocytes in the perinatal period is largely unknown. Our results demonstrate that Nr4a1 expression is highest among Ly6C^Lo^ non-classical monocytes in the neonatal liver under homeostatic and inflammatory conditions. Despite the decrease in the proportion of Ly6C^Lo^ non-classical monocytes in the liver during inflammation, the intensity of Nr4a1 expression in this pro-reparative subset significantly increased in response to inflammation. These data suggest that a small population of Nr4a1^Hi^Ly6C^Lo^ monocytes may play a functional role during murine BA. Moreover, our data suggest that Nr4a1 may play a role in the regulation of both Ly6C^Lo^ monocytes and macrophages during inflammation. Though further work will be needed to specifically define the separate roles of Ly6C^Lo^ monocytes and macrophages, our data highlight the potential implications of using Nr4a1 as a regulator of neonatal hepatic Ly6C^Lo^ non-classical monocytes to mitigate inflammatory injuries.

Further investigation is needed to determine the functional significance of Nr4a1 in myeloid immune responses during perinatal liver inflammation. Previous studies have shown that Nr4a1-deficient mice have significantly fewer Ly6C^Lo^ non-classical monocytes circulating in their blood or spleen, or patrolling the endothelium [13]. Moreover, the few Ly6C^Lo^ non-classical monocytes present in the bone marrow of Nr4a1-deficient mice were found to be arrested in the S phase of the cell cycle, and rapidly underwent apoptosis [13]. Given that the patterns of Nr4a1 expression in neonatal pups in our study are similar to those in studies of adult bone marrow [13], we anticipate that neonatal mice deficient in Nr4a1 will also have significantly fewer Ly6C^Lo^ non-classical monocytes, and will therefore not be able to mitigate perinatal liver inflammation effectively. Further studies of the relationship between Nr4a1 and pro-reparative Ly6C^Lo^ non-classical monocytes are needed before potential therapeutic targets can be developed to halt BA and other inflammatory diseases.

## 5. Conclusions

In conclusion, our results demonstrate that the transcription factor Nr4a1 is expressed at its highest level in Ly6C^Lo^ non-classical monocytes in the neonatal mouse liver. Furthermore, in response to inflammation, these levels decreased in Ly6C^Lo^ non-classical monocytes in the bone marrow, but increased in Ly6C^Lo^ non-classical monocytes in the liver. These findings highlight the potential implications of using Nr4a1 as a regulator of neonatal hepatic Ly6C^Lo^ non-classical monocytes to mitigate perinatal liver inflammation.

## Figures and Tables

**Figure 1 jcm-11-05290-f001:**
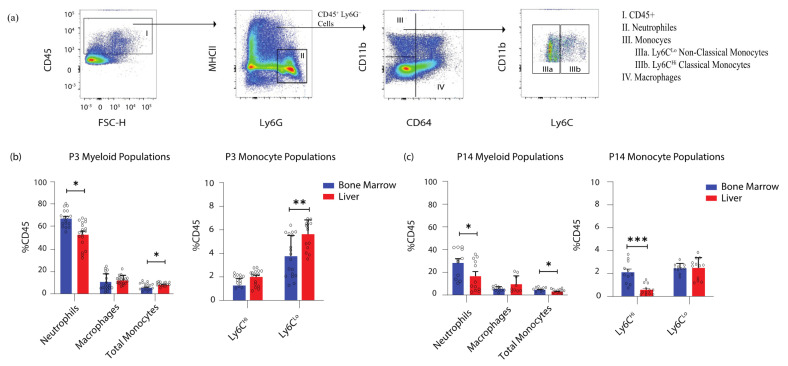
**Pro-reparative Ly6C^Lo^ non-classical monocytes reside in the liver at the early neonatal timepoint.** (**a**) Myeloid populations including neutrophils, macrophages, Ly6C^Hi^ classical monocytes and Ly6C^Lo^ non-classical monocytes identified by flow cytometry. (**b**) Myeloid populations at an early neonatal timepoint (P3) as a proportion of all CD45% leukocytes. The neonatal liver (*n* = 19) harvests significantly more monocytes, particularly Ly6C^Lo^ non-classical monocytes, compared to the BM (*n* = 18) (BM 3.76% vs liver 5.62%; *p* = 0.0015). (**c**) At the juvenile timepoint (P14) the liver (*n* = 12) no longer harvests monocytic populations. Ly6C^Hi^ classical monocytes predominately reside in the BM (*n* = 11) at this point (BM 2.11% vs liver 0.58%, *p* < 0.0001). * *p* ≤ 0.05; ** *p* ≤ 0.01; *** *p* ≤ 0.001.

**Figure 2 jcm-11-05290-f002:**
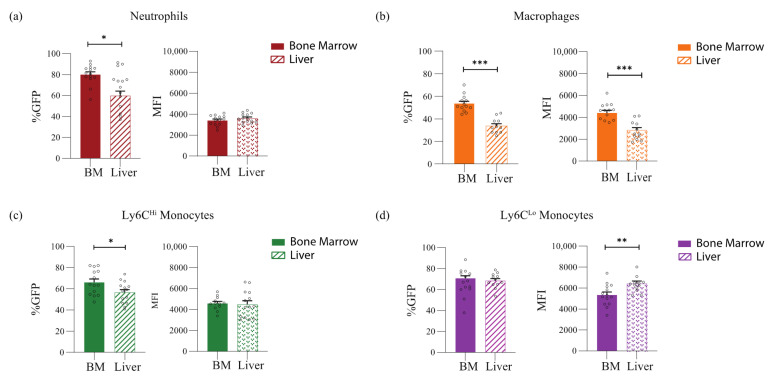
**Ly6C^Lo^ non-classical monocytes exhibit the highest expression of Nr4a1.** (**a**) Nr4a1 expression on neutrophils as %GFP positive (*n* = 14; BM 80.03% vs liver 66.52%; *p* = 0.02). Both organs exhibit similar intensity of expression, represented as Mean Fluorescence Intensity (MFI) (BM 3408 vs liver 3611; *p* = 0.25). (**b**) Neonatal BM macrophages show higher expression of Nr4a1 compared to the liver (*n* = 14; BM %GFP 53.46% vs liver 33.83; *p* < 0.001) (BM MFI 4554 vs liver 2782; *p* < 0.0001). (**c**) Higher proportion of Ly6CHi classical monocytes exhibit Nr4a1 expression in the neonatal BM (*n* = 14; BM 66.13% vs Liver 56.51; *p* = 0.03); however, MFI is similar between both organs (BM 4575 vs liver 4485; *p* = 0.82). (**d**) High proportions of Ly6C^Lo^ non-classical monocytes exhibit Nr4a1 expression in both organs (*n* = 14; BM 66.60 vs liver 68.42%; *p* = 0.67); however, Ly6C^Lo^ non-classical monocytes in the neonatal liver show a higher intensity of expression (BM MFI 5338 vs liver 6344; *p* = 0.009). (**e**) Among the myeloid populations in the neonatal liver, Ly6C^Lo^ non-classical monocytes exhibit the highest expression of Nr4a1 compared to neutrophils (MFI 6344 vs 3611, respectively; *p* < 0.0001), macrophages (MFI 6344 vs 2782, respectively; *p* < 0.001), and Ly6C^Hi^ classical monocytes (MFI 6344 vs 4485, respectively; *p* < 0.0002). (**f**) Ly6C^Lo^ non-classical monocytes also exhibit the highest expression of Nr4a1 in the neonatal BM; however, the degree of intensity is significantly lower than in the liver. * *p* ≤ 0.05; ** *p* ≤ 0.01; *** *p* ≤ 0.001.

**Figure 3 jcm-11-05290-f003:**
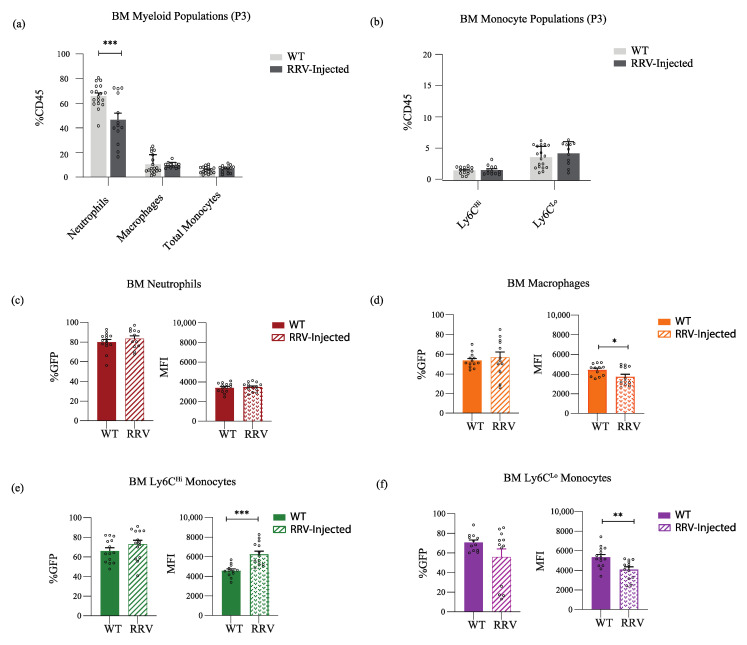
**BM Ly6C^Lo^ non-classical monocytes decrease their Nr4a1 expression in the setting of inflammation.** (**a**,**b**) Myeloid populations and monocyte subsets at (P3) as a proportion of all CD45% leukocytes under homeostatic (*n* = 18) and inflammatory (*n* = 14) conditions in the neonatal BM. (**c**–**f**) %GFP and intensity of Nr4a1 expression (MFI) on myeloid populations in the BM under homeostatic conditions (*n* = 14) and in the setting of inflammation (*n* = 14). * *p* ≤ 0.05; ** *p* ≤ 0.01; *** *p* ≤ 0.001.

**Figure 4 jcm-11-05290-f004:**
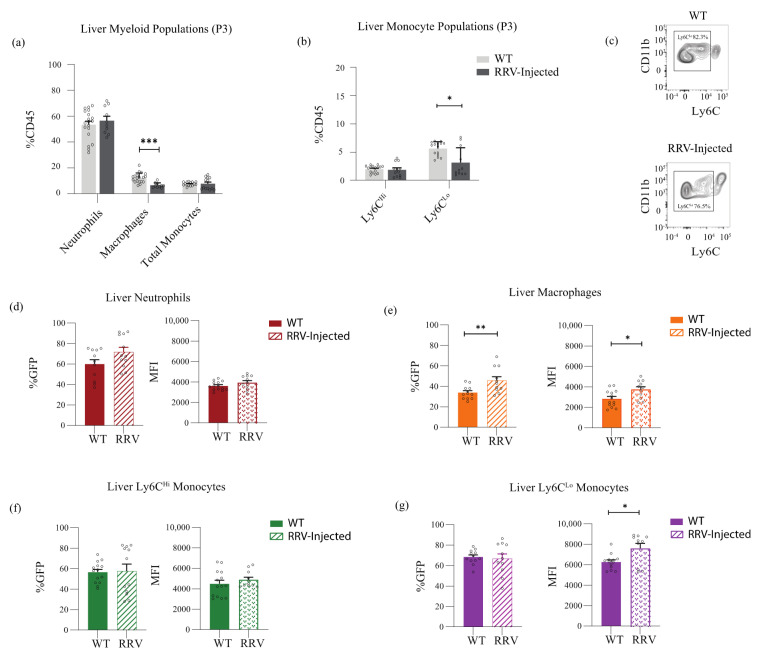
Ly6C^Lo^ non-classical monocytes increase Nr4a1 expression in the setting of inflammation. (**a**,**b**) Myeloid populations and monocyte subsets at (P3) as a proportion of all CD45% leukocytes under homeostatic (*n* = 19) and inflammatory (*n* = 13) conditions in the neonatal liver. (**c**) Reduction of Ly6C^Lo^ non-classical monocyte population in the liver by flow cytometry. (**d**–**g**) Intensity of Nr4a1 expression (MFI) is increased on macrophages and Ly6C^Lo^ non-classical monocytes in the setting of inflammation (WT *n* = 14, RRV *n* = 12). * *p* ≤ 0.05; ** *p* ≤ 0.01; *** *p* ≤ 0.001.

## Data Availability

The data presented in this study are available on request from the corresponding author.

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
