# Peer review of "Hepatic Ly6CLo Non-Classical Monocytes Have Increased Nr4a1 (Nur77) in Murine Biliary Atresia"

_jcm, 2022, doi:10.3390/jcm11185290_

Round 1

Reviewer 1 Report

This is a experimental study evaluating Nr4a1 expression in hepatic monocytes. I don't understand the importance of Nr41 in murine biliary atresia. Did the authors use model of biliary atresia?

Author Response

  1. This is a experimental study evaluating Nr4a1 expression in hepatic monocytes. I don't understand the importance of Nr41 in murine biliary atresia. Did the authors use model of biliary atresia?

We thank the Reviewer for this comment and apologize that the importance of Nr4a1 in biliary atresia (BA) was not made clear. As mentioned in the Introduction, our previous work implicates non-classical monocytes in the pathogenesis of perinatal liver inflammation in a murine model of BA (Alkhani, Levy et al. 2020). Based on the importance of Nr4a1 for non-classical monocyte survival and differentiation, we studied the effects of inflammation on Nr4a1 expression in a murine model of BA. We have clarified this point in both the Introduction and Discussion.

Furthermore, additional work will be needed to specifically prove that the pro-reparative functions exhibited by non-classical monocytes in murine BA are Nr4a1 dependent. This caveat of our study has been included in the Discussion (Line 293).

Reviewer 3 Report

The manuscript entitled “Hepatic Ly6CLo Non-Classical Monocytes Have Increased Nr4a1 (Nur77) in Murine Biliary Atresia” involved an interesting question, but there are some questions that need to be resolved by the authors before it is ready:

Methodology
Please inform the age, weight, and other animals characteristic. Also, inform the methodology of euthanasia. The numbers of animals used per group (and the groups of analysis made it)?
For single cells-suspensions, made the authors liver perfusion or another protocol to ensure cleaned tissue before cell obtention? For erythrocytes or other blood contaminates, did the authors use any type of lysis buffer? The methodology is not clear or informative. Please correct it. Also, how do the authors determine the percentage of cell survivance?
In line 83 “the pelvis, sacrum, femur, and tibia 82 of both P3 and P14 animals were harvested and homogenized by mechanical dissociation” what was the cell extraction protocol?

In discussion, figure 4G (%GFP) how do the authors explain the similar quantities of Ly6Clo in WT and RRV? And discuss the difference between this and the enhancement observed by the macrophagic population.
For to ensure that Nr4a1 is differential (tissue-specific) the authors must make a WB of Nr4a1 (nuclear extract will be ideal for explaining the tissue-specific changes).

Round 2

Reviewer 1 Report

The authors have revised the manuscript appropriately.

Reviewer 3 Report

The authors made all the suggested requests and the manuscript looks well.